# Parasitoid Wasp *Acerophagus papayae*: A Promising Solution for the Control of Papaya Mealybug *Paracoccus marginatus* in Cassava Fields in Vietnam

**DOI:** 10.3390/insects14060528

**Published:** 2023-06-06

**Authors:** Khac Hoang Le, Thi Hoang Dong Tran, Dang Hoa Tran, Tuan Dat Nguyen, Cong Van Doan

**Affiliations:** 1Entomology Laboratory, Department of Plant Protection, Faculty of Agronomy, Nong Lam University Ho Chi Minh, Ho Chi Minh 700000, Vietnam; lkhoang@hcmuaf.edu.vn (K.H.L.);; 2Faculty of Agronomy, University of Agriculture and Forestry, Hue University, Hue City 530000, Vietnam; 3The Plant Physiology Unit, The Department of Life Sciences and Systems Biology of the University of Turin, Via Accademia Albertina 13/Via Gioacchino Quarello 15/A, 10123 Torino, Italy

**Keywords:** parasitoid wasp, mealybugs, *Acerophagus papayae*, *Paracoccus marginatus*, cassava, biocontrol

## Abstract

**Simple Summary:**

This study examined the occurrence, biological characteristics, and effectiveness of the parasitoid wasp *Acerophagus papayae* Noyes and Schauff in controlling the papaya mealybug *Paracoccus marginatus* Williams and Granara de Willink, a major pest of cassava crops in Vietnam. The results showed that *A. papayae* naturally occurred more frequently than other species of parasitoid wasps. The honey solution was found to be important for increasing the lifespan of *A. papayae* in the absence of hosts. The second instar of the mealybug was found to be the most suitable host stage for parasitism by *A. papayae*. Female wasps can produce a large number of eggs during their entire lives. These findings suggest that *A. papayae* could be an effective agent for controlling *P. marginatus* and improving crop pest management not only in Vietnam, but in other regions as well.

**Abstract:**

Cassava is a valuable export commodity crop that is often attacked by pests, causing economic losses for this crop. The papaya mealybug *Paracoccus marginatus* has become a major pest of cassava in Vietnam. The parasitoid wasp *Acerophagus papayae* has been demonstrated to be the most efficient parasitoid wasp for controlling *P. marginatus* in many regions. We observed the occurrence of *A. papayae* in Vietnam, studied the biological characteristics of *A. papayae*, and investigated its parasitic activity on *P. marginatus*. The results showed that *A. papayae* occurred more frequently than *Anagyrus loecki*, another known parasitoid of *P. marginatus*. The lifespan of *A. papayae* was approximately 16 days. In the absence of hosts, a 50% honey solution was an essential diet to increase the longevity of both female and male of *A. papayae*. The second instar of *P. marginatus* was a suitable host stage for parasitism by *A. papayae*. Female *A. papayae* laid approximately 60.8 eggs within 17 days, mostly during the first 6 to 7 days. These findings suggest that *A. papayae* has the potential to control *P. marginatus*, and could inform the development of more effective pest management strategies for cassava crops in Vietnam and other regions affected by this pest.

## 1. Introduction

Cassava (*Manihot esculenta* Crantz) is a woody shrub in the Euphorbiaceae family, primarily cultivated for its starchy roots. It is widely grown in tropical and subtropical regions of Africa, Asia, and Latin America, with a global production of over 302 million tons per year [1]. Cassava is a staple food for over 500 million people worldwide, particularly across Africa (with 63% of total world production), where it is considered key in combatting food security issues [2,3,4]. Cassava is also an important export commodity for processing into products such as monosodium glutamate, alcohol, confectioneries, instant noodles, plywood, packaging, biofilm, and pharmaceutical additives. Moreover, cassava is an essential fodder crop in many countries [5,6,7]. In Vietnam, cassava was introduced in the early 19th century, and has since become a high-value crop, contributing significantly to poverty reduction for farmers in upland, remote, and isolated areas [6,8,9].

The growth period of cassava varies from 6 to 12 months, depending on the variety, growing season, growing area, and intended use [8,10,11]. Cassava plants are vulnerable to numerous pests [12,13]. In South America, nearly 200 species of arthropods feed on cassava as their primary food source, resulting in yield losses of over 50% [14]. In Vietnam, the insect pests of cassava have not been thoroughly studied, and only a few harmful arthropod species have been identified [15]. Mealybugs are insects that cause damage to cassava, including *Ferrisia virgata* Cockerell (Hemiptera: Pseudococcidae), *Paracoccus marginatus* Williams and Granara de Willink (Hemiptera: Pseudococcidae), *Phenacoccus manihoti* Matile-Ferrero (Hemiptera: Pseudococcidae), and *Pseudococcus jackbeardsleyi* Gimpel and Miller (Hemiptera: Pseudococcidae) [16]. In Vietnam, *P. marginatus* was first discovered in Tay Ninh province in 2012, and has since become the primary pest of cassava in 14 provinces and cities throughout the country [16]. According to Williams and Willink [17], *P. marginatus* is native to Belize, Costa Rica, Guatemala, and Mexico, and has spread to some Caribbean islands, Hawaii, and French Guiana [18]. *Paracoccus marginatus* was first discovered on hibiscus plants in Bradenton, Florida, USA, in 1998, and was subsequently found on 18 different plant species in the state [19].

*Paracoccus marginatus* attacks more than 55 species of host plants, including high-value crops such as papaya, avocado, citrus, mango, strawberry, and pomegranate, as well as hibiscus, cotton, tomato, eggplant, peppers, beans, peas, sweet potatoes, and tapioca [18,19,20,21,22]. This pest is omnivorous and feeds by sucking sap through the epidermis of leaves, fruits, and stems, which leads to leaf wrinkling, yellowing, and wilting, eventually causing plant death [23]. In addition, *P. marginatus* secretes black molds that cover leaves, fruits, and stems, impeding photosynthesis and gas exchange, resulting in stunted growth, deformed leaves, and, ultimately, tree death [21,23,24]. Since it was first reported in Coimbatore in 2007, *P. marginatus* has been identified as a key pest causing severe economic losses to farmers in Coimbatore, Erode, Tirupur, and Salem in India [23]. For example, strawberry farmers suffered losses of over 1500 hectares in Tirupur in 2009, resulting in heavy financial losses [23]. Potential economic losses caused by *P. marginatus* can total between 57% and 75% on papaya trees, with a decrease in fruit quality and a 2–3-fold increase in the cost of production [25,26].

For the long-term management of *P. marginatus*, biological control is the most sustainable and cost-effective option [15,27]. Natural enemies of mealybugs, such as the Australian ladybug *Cryptolaemus montrouzieri* Mulsant (Coleoptera: Coccinellidae), *Scymnus* Kugelann (Coleoptera: Coccinellidae), and some parasitoid wasps [23,27], are present in nature. *Acerophagus papayae* Noyes and Schauff, *Anagyrus loecki* Noyes and Menezes, and *Pseudleptomastix mexicana* Noyes and Schauff (Hymenoptera: Encyrtidae) were regularly introduced to control *P. marginatus,* especially in Guam, Palau, Florida and Hawaii (USA), India, and Sri Lanka [22,24,28,29]. Among the parasitoid wasps, *A. papayae* (Hymenoptera: Encyrtidae) is one of the most effective species in controlling *P. marginatus* [30,31,32,33]. Mastoi et al. [34] reported on the natural enemies of *P. marginatus* in the Malaysian Peninsula, and observed *A. papayae* to be a parasitoid wasp on this mealybug. The incidence of mealybugs significantly decreased since the introduction of *A. papayae* into the environment [32,35]. Meyerdirk et al. [36] reported that 46,200 individuals of the parasitoid wasps *A. loecki*, *P. mexicana*, and *A. papayae* had been introduced from Puerto Rico and released in Guam from June to October 2002, resulting in a reduction of more than 99% of *P. marginatus* abundance after 1 year. This has reduced the risk of invasion of *P. marginatus* to neighboring islands. The effectiveness of three parasitoid wasp species (*A. loecki*, *P. mexicana*, and *A. papayae*) against *P. marginatus* was observed in the 4 months after release of the parasitoid wasps in an ecological mulberry garden in Tamil Nadu, India. *Acerophagus papayae* exhibited the highest population growth and parasitism rate, of 75.6–81.7%, followed by *P. mexicana* with 9.3–24.4%; the lowest parasitism was exhibited by *A. loecki* of 0.7–9.0% [37].

Although *P. marginatus* is becoming a key pest, damaging cassava in Vietnam, there have not been many studies evaluating the existence of parasitoid wasps to control this pest in various conditions [27]. Therefore, this study was conducted to (1) determine and identify the presence of the parasitoid wasp *A. papayae* in cassava fields in Vietnam; (2) determine the life cycle of *A. papayae* under specific conditions; and (3) determine the parasitism of *A. papayae* on *P. marginatus* in Vietnam, including its preferred host stage and egg-laying rhythm.

## 2. Materials and Methods

### 2.1. Study Sites and Biological Materials

The experiment was conducted during the 2016–2017 cropping season at two locations. The first location was Cam Giang commune, Go Dau district, Tay Ninh province, Vietnam (11°16′10.385′′ N and 106°5′10.96′′ E), where the frequency of occurrence of the parasitoid wasp *A. papayae* and the papaya mealybug *P. marginatus* were investigated. The second location was the insect laboratory at the Department of Plant Protection, Department of Agronomy, Nong Lam University, Ho Chi Minh City, Vietnam (10°52′17.126′′ N and 106°47′31.423′′ E), where all the experiments were conducted.

Rearing of *P. marginatus*: *P. marginatus* were collected from cassava fields and reared on a cassava plant in a net cage. When *P. marginatus* laid eggs, all eggs of the same age were transferred to new cassava plants to ensure a source of similarly aged mealybugs for experiments.

Rearing of parasitoid wasps *A. papayae*: Adult male and female specimens of *A. papayae* obtained from the cassava field were paired in collection tubes and provided with a 50% honey + 50% water solution (5–10 pairs of wasps exposed to second-instar *P. marginatus* on cassava in the net cage). After 24 h, the cassava plants with mealybugs were separated and kept in the net cage until *A. papayae* emerged. Emerging adults from infested mealybugs were used to reproduce wasps for the experiments.

### 2.2. Experimental Designs

#### 2.2.1. Biological Characteristics of *A. papayae*

##### Determining the Frequency of Parasitoids of the *P. marginatus* in Vietnam

The occurrence of *P. marginatus* parasitoids was recorded once a week during the cassava season. Parasitoids were collected from three cassava fields with high *P. marginatus* damage (>30%) and an area of 0.5–1 ha. Five collection points were randomly selected along two diagonal lines. At each collection point, five adjacent cassava bushes were selected. We cut off the *P. marginatus*-infested part of the cassava plant (randomly picking four points on each tree) and placed them in a net cage to collect the parasitoid wasps. The wasps were identified using Noyes and Schauff’s method [38].

The occurrence frequency of parasitoid wasps was assessed as follows:
 −  Very low  <5%  +  Low  5–25%  ++  Moderate  25–50%  +++  High  50–75%  ++++  Very high  >75% 

Relative occurrence (%) = (*f*/*n*) × 100, where f is the number of individual cassava plants having parasitoid wasps of the papaya mealybug, and n is the total number of cassava observations.

Observing the Mating Activity and Egg-Laying Behavior of *A. papayae* Under Laboratory Conditions

The experiment aimed to observe the mating activity and egg-laying behaviors of *A. papayae* on *P. marginatus*. Ten pairs of newly emerged *A. papayae* were placed in a collection tube and provided with two hundred second-instar *P. marginatus* on cassava leaves. The mating activity and egg-laying behavior of *A. papayae* were observed under a stereo microscope (Olympus SZ51—Japan, 40× magnification). The experiments were repeated for seven days to record the common mating activity and egg-laying behavior of *A. papayae* on *P. marginatus*.

##### Determining the Immature Developmental Time of the Parasitoid Wasps *A. papayae*

The experiments aimed to characterize the immature developmental time of the parasitoid wasps *A. papayae*, which can inform the optimal timing for releasing the wasps into the environment to control mealybugs. Ten pairs of *A. papayae* (fed with 50% honey) were exposed to 200 individuals of *P. marginatus*, which were fed papaya fruit in a net cage. After 24 h of exposure, the papaya fruit was removed from the cage and placed in another net cage. Every day, ten specimens were dissected and observed under a stereo microscope and electron microscope to determine the life cycle of the wasp. The life cycle of the *A. papayae* was determined from the *P. marginatus* parasitism stage to the adult stage [39].

##### Effect of Diets on the Longevity of Adult *A. papayae*

In order to provide a diet for *A. papayae* in the absence of hosts, we tested the effect of diets (honey concentration) on the longevity of the parasitoid adults. The experiment was arranged in a completely randomized single-factor design with five treatments (control—distilled water, 10% honey, 30% honey, 50% honey, and 70% honey) with ten replicates. Each replicate included one adult *A. papayae* wasp in a test tube, with the end of test tube covered with a mesh, and cotton wool soaked in honey. After 24 h, the cotton wool was refreshed until the wasp died. The longevity of the parasitoid wasp was recorded as the average lifespan of adult *A. papayae* wasp per treatment.

#### 2.2.2. Parasitism of *A. papayae* on *P. marginatus*

##### Host Stage Susceptibility and Preference of *A. papayae*

This experiment aimed to identify the host stage susceptibility and preference of *A. papayae*, establish an efficient method for the mass rearing of *A. papayae*, and determine the efficiency of controlling *P. marginatus* under natural conditions. The experiment was conducted in a completely randomized single-factor design with four treatments (second-instar nymphs, third-instar nymphs, non-laying-eggs adults of *P. marginatus*, and egg-laying adults of *P. marginatus*), each with 10 replicates. Each replicate included a female wasp that had mated for three days and was fed with a 50% honey solution (the most optimal concentration of the honey solution to rearing the parasitoid). The female was exposed to 20 *P. marginatus* individuals in each stage. After 24 h of exposure, each treatment was assessed using the following parameters [39]:Parasitism rate (%) = (number of infected *P. marginatus*/total *P. marginatus*) × 100
Emergence rate (%) = (number of *A. papayae* emerging/total number of infected *P. marginatus*) × 100
Female rate (%) = (number of emerging female/total number of emerging) × 100

##### Parasitoid Fecundity and Progeny of *A. papayae*

The aim of this experiment was to determine the total number of eggs laid by *A. papayae* on *P. marginatus* in the second-instar nymph. One pair of newly emerged *A. papayae* was fed on 50% honey and exposed to 20 *P. marginatus* individuals in the second-instar stage. After 24 h, *P. marginatus* and honey were replaced under the same conditions. Twenty of the exposed *P. marginatus* were dissected to determine the number of eggs laid in the body of the mealybug, as well as the egg-laying ability of *A. papayae*. The experiment was continued until the parasitoid wasps died. The observations were repeated 10 times, each time with a new pair of *A. papayae* individuals. The observation included the total number of eggs laid by one female wasp and the number of eggs laid per day.

##### The Effect of Parasitism Activity on the Longevity of Female *A. papaya*

The experiment aimed to determine the effect of parasitism activity on the longevity of the female *A. papayae*. The study followed a completely randomized single-factor design with two treatments (nonparasitism activity and parasitism activity), including 10 replicates. Each replicate consisted of a paired *A. papayae* that was fed with 50% honey within 24 h. Then, the wasp was either placed in a test tube without *P. marginatus* + 50% honey or with the second instar of *P. marginatus* and 50% honey. New 50% honey was replaced every 24 h until the parasitoid wasps died. The average longevity of female parasitoid wasp (days) was recorded by observing the number of female wasp deaths every 24 h.

### 2.3. Data Analysis

Data were entered, processed, and graphed using Excel (2020). For data analysis, the proportional data, such as the parasitism rate (%), emergence (%), and female emergence (%) data, were transformed by arcsin√(%) to meet normality [40,41,42] before analysis. The response variables, such as the parasitism rate, the emergence, the female emergence, and the longevity of female and male *A. papayae,* were assessed by ANOVA analysis of variance IBM SPSS Statistics 28.0 software with the main effect of different host development stages or honey solutions. In addition, Tukey’s post hoc test was performed to separate the means at the 95% confidence interval, where ANOVA showed significant differences in the means. The Student’s *t*-test was employed to analyze the effect of parasitism activity on the longevity of female *A. papayae*.

## 3. Results

### 3.1. Diversity and Abundance of P. marginatus Parasitoids in Vietnam

Only two species of parasitoid wasps belonging to the Encyrtidae family, *Acerophagus papayae* and *Anagyrus loecki*, were recorded on *P. marginatus* in Cam Giang commune, Go Dau district, Tay Ninh province, Vietnam. *A. papayae* occurred at a high frequency of 53.9%, whereas *A. loecki* was observed with a moderate frequency of 34.9% (Table 1).

### 3.2. Mating Activity and Egg-Laying Behavior of A. papayae under Laboratory Conditions

During the mummy stage of the mealybug, the head of *A. papayae* emerges first, and the body segments stretch from the back to the front until the body completely exits the mummy. After emerging, female *A. papayae* can mate and lay eggs within 24 h. The mating process starts when the male courts the female wasp. If the female accepts, the female stands 3 to 4 cm away from the male wasp, moving her antennae continuously and raising her wings slowly. Next, the male jumps onto the back of the female and clings to her back with his feet. The male wasp then bends the end of his abdomen and inserts his genitals into the female wasp’s genitals. Mating occurs for approximately 10 s, during which the female remains still. At the end of mating, the male is pushed 4 to 5 cm away by the female. The male stands motionless for 15 to 20 s, then fans his wings to dry the mating parts, and proceeds to pair and mate with other female wasps. If the mating process fails, the male will jump onto the back of the female and mate again.

Female *A. papayae* usually lay eggs in areas where there are many *P. marginatus*. When searching for a host, the wasp’s antennae reach forward, similar to two “antennas”, to find the host’s scent, locating a potential host more quickly. When preparing to lay eggs, the female wasp crawls onto the host’s back, finds a suitable location, and uses the egg-laying tray to puncture the skin and lay eggs in the host’s body. The egg-laying time of females ranges from 25 to 30 s. After oviposition, the female leaves the parasitized host without any action to mark the host as parasitized. Therefore, if *A. papayae* chooses an already-parasitized host, the *P. marginatus* will respond by flexing its body, rapidly expanding its abdomen to prevent the female from laying eggs.

It was observed that the female *A. papayae* had a habit of feeding on the host’s body fluid in addition to laying eggs. The female wasp pierces the skin on the host’s dorsal side with a trough, and then turns her head to use her mouth to extract fluids from the hole in the skin. This causes the mealybug’s body to become parasitized, mummified, and die. Thus, *A. papayae* is considered to have biological potential for managing *P. marginatus* populations.

### 3.3. Life Cycle of the A. papayae and Developmental Time of Each Phase

The maturation time of the egg was 3.7 ± 0.16 days. The larvae developed in three different stages with two molt processes. The first, second, and third instars had maturation times of 1 to 2 days between each stage. The total maturation time of the larval phase was 4.2 ± 0.22 days. At the end of the third instar stage, *A. papayae* molted for the last time to move into the prepupal phase, which lasted 2.7 ± 0.16 days. After the prepupal phase, *A. papayae* progressed into the pupal phase; the pupal phase was 8.0 ± 0.28 days. The total maturation time of the pupa (including prepupal and pupal phases) was 10.0 ± 0.28 days. The life cycle of *A. papayae* was completed in 16.0 ± 0.32 days, from egg to adult emergence (Appendix A).

### 3.4. The Potential of Using Honey Solution as a Diet for A. papayae in the Absence of a Host

The 50% honey diet solution influenced the longevity of both male (*p* ≤ 0.01, *F* = 28.089) and female *A. papayae* (*p* ≤ 0.01, *F* = 78.115). Adult female and male *A. papayae* fed on 50% honey survived for 12.5 ± 0.40 days and 7.3 ± 0.34 days, respectively. Longevity was shortest for those fed on distilled water: 3.2 ± 0.13 and 2.2 ± 0.25 days for females and males, respectively (Table 2).

### 3.5. Susceptibility and Preference of Different Host Stages for A. papayae

The parasitism rates (%) of *A. papayae* on different host stages of *P. marginatus* were significantly different (*p* ≤ 0.01, *F* = 143.240). The highest parasitism rate of 46% was observed when *A. papayae* parasitized the second-instar stage of *P. marginatus*. In contrast, no parasitism was observed on egg-laying adults (Figure 1).

Similarly, the emergence values (%) of *A. papayae* from different host stages of *P. marginatus* were significantly different (*p* ≤ 0.01, *F* = 73.853). The emergence of *A. papayae* was high on second-instar, third-instar, and non-egg-laying adults. The emergence ranged from 86% to 89%, compared with 0% for egg-laying adults (Figure 2).

The numbers of female emergences (%) of *A. papayae* parasitizing different stages of *P. marginatus* were significantly different (*p* ≤ 0.01, *F* = 142.741). The lowest female emergence rate was observed on egg-laying adults (0%), while the highest female emergence was 74% on non-egg-laying adults and third instars. The second instar showed the most balanced gender ratio of emerging *A. papayae* (Figure 3).

### 3.6. Acerophagus papayae Reproduction on P. marginatus

Female *A. papayae* laid approximately 60.8 ± 1.17 eggs during their life (Appendix A). The egg-laying rhythm of *A. papayae* decreased with each day. The female wasp laid eggs over 17 days, of which the most was 10 ± 0.54 eggs on the first day; and the highest concentration was laid in the first 6 to 7 days after starting oviposition (Figure 4).

### 3.7. Parasitism and Longevity of Female A. papayae

Female wasps that exhibited parasitism activity lived longer than the females without parasitoid activity. Females that participated in parasitism activities had an average longevity of 14.2 ± 0.59 days, whereas wasps without any parasitism activity had an average longevity of 12.5 ± 0.40 days (Table 3).

## 4. Discussion

Tay Ninh province was reported to be an invasion area of mealybugs, which cause damage to cassava. During field surveys conducted from 2014 to 2015, four different mealybug species, *Paracoccus marginatus*, *Phenacoccus manihoti*, *Pseudococcus jackbeardsleyi*, and *Ferrisia virgata*, were recorded in cassava fields. The abundance of mealybugs ranged from 0 to 698 per cassava tip [27]. *Paracoccus marginatus* is widely expanding across Central and East Africa, Central America, and Asia, because these areas have suitable climatic conditions and diverse host crops [22,43]. The presence of *P. marginatus* may increase the presence of the natural enemies in the field; however, the abundance is normally too low for controlling this pest [34,36]. *Anagyrus lopezi* was found to be parasitizing *P. marginatus* in Tay Ninh province. However, the level of parasitism was not efficient to control *P. marginatus* [27]. In this study, we not only observed *A. loecki*, but also *A. papayae*, parasitizing *P. marginatus* (Table 1). *A. papayae* has been successfully used as biocontrol agent in different regions around the world [36,37]. Our findings indicate a potential new addition to the list of parasitoid wasp species which attack *P. marginatus* in Vietnam [16,24,27,30,31,32]. However, the abundance of these parasitoids was not efficient for the biocontrol of *P. marginatus* [43]. Consequently, more wasps need to be mass-reared to mitigate against the increasing damage caused by *P. marginatus* to cassava [36,37].

*Acerophagus papayae* has the potential to be mass-produced for biocontrol programs [44]. To effectively rear *A. papayae* for biocontrol strategies, it is crucial to have a good understanding of their immature developmental time, longevity, and lifetime fertility [45]. Each female parasitoid wasp can produce approximately 60 eggs in total over their lifespan (Figure 4). Female *A. papayae* lay most of their eggs within the first 6 to 7 days after emergence, which is a critical stage to consider when determining the optimal timing to release the wasp into the wild. This enables the wasp to adapt to the natural environment and take full advantage of their egg-laying potential [45].

Determining the appropriate host age is crucial for the efficient multiplication of parasitoid wasp biomass and estimating the optimal time to release the wasps into the environment [39,46]. The preference for a particular host stage is vital for the viability of young *A. papayae*. Adequate and appropriate nutrition are necessary to ensure the complete life cycle development of the parasitoid wasp [47]. The host must have an appropriate body size and sufficient time for the parasitoid wasp to complete its life cycle [48]. *A. papayae* parasitizes *P. marginatus*, and the host responds by stretching its body, which leads to the parasitoid wasp breaking its egg-laying tube and dying. The parasitism rate gradually decreases with the age of the host; older hosts exhibit a stronger response. The second instar of *P. marginatus* is a suitable host for *A. papayae*, with high emergence rates and a well-balanced sex ratio (Figure 1 and Figure 3). This is important for multiplying parasitoid wasp numbers and determining the time to undertake field releases to manage the target pest [39,45]. The *A. papayae* wasp only requires a single *P. marginatus* at the second-instar nymph stage to develop from an egg to adult (See Appendix A).

*A. papayae* fed with a honey solution had a significantly higher longevity than when fed with distilled water, as previously described [49,50]. This potentially increases the longevity of female *A. papayae* in the absence of a host. Females exhibiting parasitoid activity had higher longevity than the females without parasitism activity [51]. Females that participated in parasitism activities had an average longevity of 14.2 days, whereas females without parasitism activities had an average longevity of 12.5 days (Table 2).

## 5. Conclusions

In Vietnam, biological measures for managing *P. marginatus* are not well studied. Our results contribute to the current knowledge about natural enemies that can be used to control *P. marginatus* in cassava cropping systems in Vietnam. To reproduce a large number of *A. papayae* for biocontrol purposes, the parasitoids can be reared on second- or third-instar nymphs of *P. marginatus* as the host. In addition, a 50% honey solution can be provided as a supplementary diet to increase the longevity of the wasps. The appropriate time to release *A. papayae* into the environment is when the majority of the *P. marginatus* population consists of second- or third-instar nymphs, and during the first 6 to 7 days after the initial egg-laying oviposition of the *A. papayae*.

## Figures and Tables

**Figure 1 insects-14-00528-f001:**
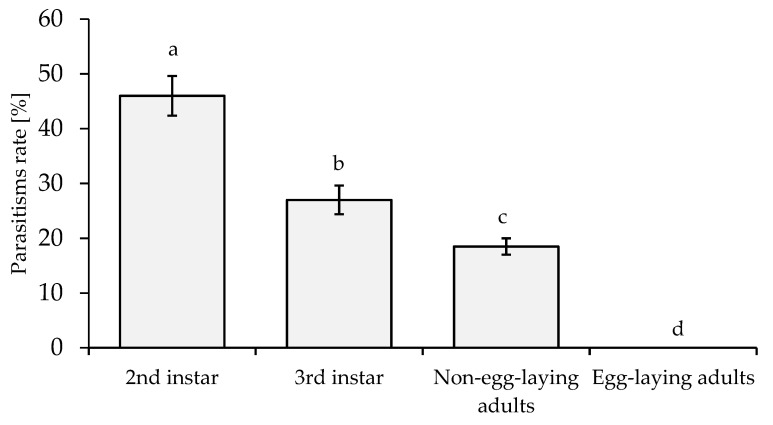
Parasitism rate (%) of *A. papayae* on *P. marginatus*. The data were transformed using arcsin√(%) function prior to statistical analysis. Different letters above columns indicate significant differences. Bar indicates SE value.

**Figure 2 insects-14-00528-f002:**
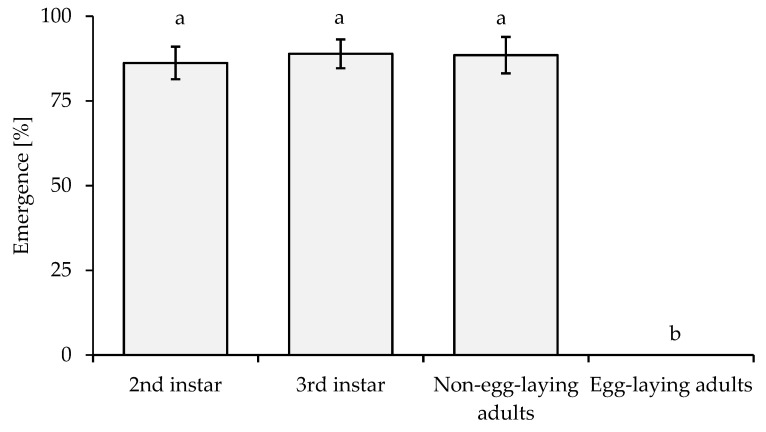
Emergence (%) of *A. papayae* on different development stages of *P. marginatus*. The data were transformed by arcsin√(%) before statistical analysis. Different letters above columns indicate significant differences. Bar indicates SE value.

**Figure 3 insects-14-00528-f003:**
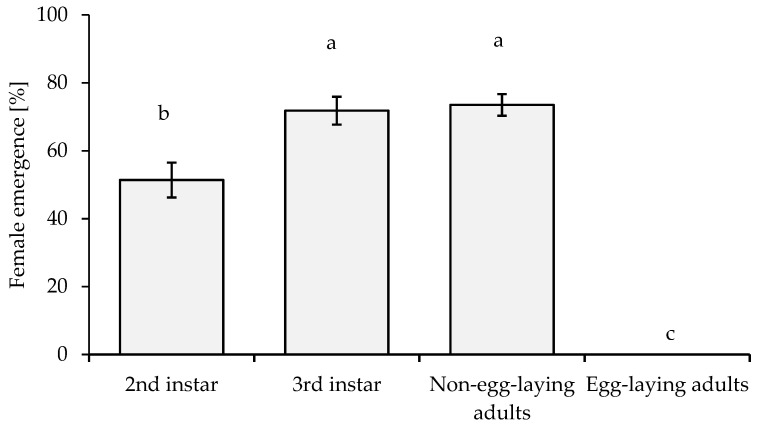
Female emergence (%) of *A. papayae* wasps parasitized on *P. marginatus*. The data were transformed using the arcsin $\sqrt{\%}$ function prior to statistical analysis. Different letters above columns indicate significant differences. Bar indicates SE value.

**Figure 4 insects-14-00528-f004:**
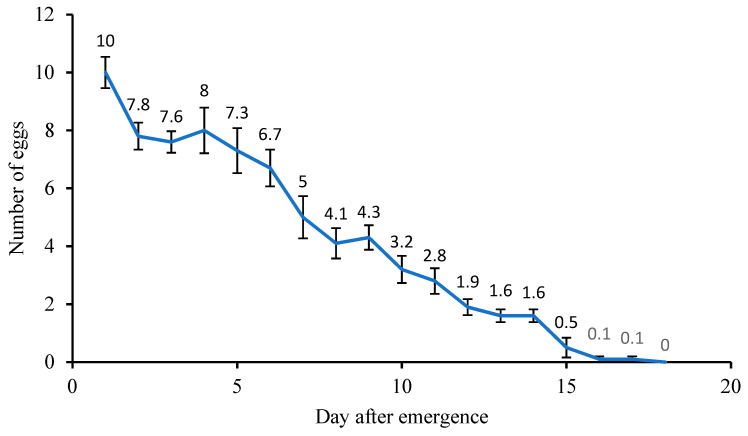
Progeny rhythm of *A. papayae*. Bar indicates SE value.

**Table 1 insects-14-00528-t001:** Occurrence frequency of parasitoid wasps of *P. marginatus* in Tay Ninh, Vietnam.

Scientific Name	Occurrence Frequency
*A. papayae*	+++
*A. loecki*	++

Note: ++, moderate (25–50%); +++, high (50–75%).

**Table 2 insects-14-00528-t002:** Effect of honey solution on the longevity of female and male *A. papayae*.

Treatments	Longevity of *A. papayae* (Day)	
	Female	Male
	(AVG ± SE)	(AVG ± SE)
Distilled water	3.2 ± 0.13 d	2.2 ± 0.25 c
Honey 10% *	6.5 ± 0.45 c	4.3 ± 0.40 b
Honey 30% *	7.6 ± 0.27 c	5.4 ± 0.43 b
Honey 50% *	12.5 ± 0.40 a	7.3 ± 0.34 a
Honey 70% *	9.6 ± 0.56 b	5.6 ± 0.34 b
*p*-value	<0.01	<0.01

Notes: AVG, average; SE: standard error; *n* = 10. Within column, means followed by the same lowercase letter do not differ significantly. *: Absence of a host.

**Table 3 insects-14-00528-t003:** Effects of parasitism activity on the longevity of female *A. papayae*.

Treatment + 50% Honey Solution	Longevity of Female *A. papayae* (Days)	
	Min–Max	AVG ± SE
Female without parasitism	11–14	12.5 ± 0.40
Female with parasitism	12–18	14.2 ± 0.59
*t*-test (*p* value)		0.0288
df (within groups)		18

Notes: AVG, average; SE, standard error; *n* = 10.

## Data Availability

The original contributions presented in this study are included in the article. Further inquiries can be directed to the corresponding authors.

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
