# Peer review of "Parasitoid Wasp Acerophagus papayae: A Promising Solution for the Control of Papaya Mealybug Paracoccus marginatus in Cassava Fields in Vietnam"

_insects, 2023, doi:10.3390/insects14060528_

Round 1
Reviewer 1 Report
This manuscript reports the biological characteristics of a parasitoid of the papaya mealybug Paracoccus marginatusinfected cassava in Vietnam. This is a routine study and the results look fine. But the analysis of data and writing need to be improved. Here are some points of consideration that I think might improve this manuscript.
1. The authors didn’t describe how they conduct the observations on of the behavior of the wasp in the method section.
2. The section titles of the results need to be consistent with those described in the method.
3. The discussion needs to be revised as I commented in the attached manuscript.
4. The common name of the mealybug and the spelling of all the scientific names of all the insect species written in the text need to check and confirm.
5. The English writhing requires improvements. Some parts of the manuscript are hard to read or misunderstandable. I have highlighted some these errors in the attached file.

The English writhing requires improvements. Some parts of the manuscript are hard to read or misunderstandable. I have highlighted some these errors in the attached file.
Author Response
Dear Reviewer,
We thank you so much for your reviewing our paper. I would like to send you back our detailed reply to your comments/suggestion. Please see the attachment
Please feel free to let us know if there is needed any further information,
Kind regards,
Doan

Reviewer 2 Report
Find comments and suggestions in the attached file

Extensive English editing is required
Author Response

(The authors gave the same response as above.)

Round 2
Reviewer 1 Report
The manuscript has been largely improved, but there are still some minor errors, for example, line 55 and 57, the name of Paracoccus marginatus (Hemiptera: Pseudococcidae) was repeated. Also, as I mentioned in the first round of review, it is preferred to use the complete name, including the author name who named the species, for all the species name when the name was shown.
The last two sentences of the discussion still need to be revised. This manuscript aims to report that Acerophagus papayae is promising, but you concluded that "these parasitoids were not efficient for the biocontrol of P. marginatus" by citing a study from India. And then you said "more wasps need to be mass-reared", again, do you mean more parasitoid species or more individuals of Acerophagus papayae?. I guess you tried to say that the natural population of Acerophagus papayae was not efficient for the biocontrol of P. marginatus, but the mass rearing of Acerophagus papayae might be efficient?
There are still some minor errors in writing, some are as mentioned above.
Author Response
Dear Reviewer,
Thank you so much for your comments on our manuscript. We improved our manuscript as your suggestion, and We would like to send you again our revised manuscript.
If you have any further comments, please feel free let us know to improve it.
Thank you so much again,
Kind regards,
Doan

Reviewer 2 Report
All my suggested edits are included in the attached

The English has greatly improved for the original submission. Additional edits are still needed as suggested.
Author Response
Dear Reviewer,
Thank you so much for your comments/detail explanation on our manuscript. We improved our manuscript as your suggestion, and We would like to send you again our revised manuscript.
If you have any further comments, please feel free let us know to improve it.
Thank you so much again,
Kind regards,
Doan

Round 3
Reviewer 2 Report
Majority of the comments have been addressed. No further comments from me.
Author Response
Dear reviewer,
Again, we thank you so much for all of your previous comments/suggestions to improve our manuscript.
Thank you for the last comments without "No further comments".
We would like to submit again our manuscript,
Kind regards,
Doan